# A Model H5N2 Vaccine Strain for Dual Protection Against H5N1 and H9N2 Avian Influenza Viruses

**DOI:** 10.3390/vaccines13010022

**Published:** 2024-12-30

**Authors:** Jin-Ha Song, Seung-Eun Son, Ho-Won Kim, Se-Hee An, Chung-Young Lee, Hyuk-Joon Kwon, Kang-Seuk Choi

**Affiliations:** 1Laboratory of Avian Diseases, College of Veterinary Medicine, Seoul National University, Seoul 08826, Republic of Korea; sjh1243@snu.ac.kr (J.-H.S.); arbre04@snu.ac.kr (S.-E.S.); iamkhw52@snu.ac.kr (H.-W.K.); 2Avian Influenza Research & Diagnostic Division, Animal and Plant Quarantine Agency, Gimcheon-si 39660, Republic of Korea; ashpri@korea.kr; 3Department of Microbiology, College of Medicine, Kyungpook National University, Daegu 41944, Republic of Korea; cylee87@knu.ac.kr; 4Research Institute for Veterinary Science, College of Veterinary Medicine, Seoul National University, Seoul 08826, Republic of Korea; 5Laboratory of Poultry Medicine, Department of Farm Animal Medicine, College of Veterinary Medicine and BK21 PLUS for Veterinary Science, Seoul National University, Seoul 88026, Republic of Korea; 6Farm Animal Clinical Training and Research Center (FACTRC), Institutes of Green Bio Science and Technology (GBST), Seoul National University, Pyeongchang 25354, Republic of Korea; 7GeNiner Inc., Seoul 08826, Republic of Korea

**Keywords:** dual protection, highly pathogenic avian influenza virus, H9N2 avian influenza virus, binary ethylenimine inactivation, N-glycosylation, NA immunity

## Abstract

**Background/Objective:** Highly pathogenic (HP) H5Nx and low-pathogenicity (LP) H9N2 avian influenza viruses (AIVs) pose global threats to the poultry industry and public health, highlighting the critical need for a dual-protective vaccine. **Methods:** In this study, we generated a model PR8-derived recombinant H5N2 vaccine strain with hemagglutinin (HA) and neuraminidase (NA) genes from clade 2.3.2.1c H5N1 and Y439-like H9N2 viruses, respectively. To enhance the immunogenicity of the recombinant H5N2 vaccine strain, N-glycans of the HA2 subunit, NA, and M2e were modified. Additionally, we replaced M2e with avian M2e to enhance the antigenic homogeneity of AIVs for better protection. We also replaced PR8 PB2 with 01310 PB2, which is the PB2 gene derived from an LP H9N2 avian influenza virus, to eliminate pathogenicity in mammals. The productivity of the model vaccine strain (rvH5N2-aM2e-vPB2) in embryonated chicken eggs (ECEs), its potential risk of mammalian infection, and the immunogenicity associated with different inactivation methods (formaldehyde (F/A) vs. binary ethyleneimine (BEI)) were evaluated. **Results:** The rvH5N2-aM2e-vPB2 strain demonstrated high productivity in ECEs and exhibited complete inhibition of replication in mammalian cells. Furthermore, compared with using F/A inactivation, inactivation using BEI significantly enhanced the immune response, particularly against NA. This enhancement resulted in increased virus neutralization titers, supporting its efficacy for dual protection against H5Nx and H9N2 avian influenza viruses. Furthermore, we demonstrated that M2e-specific immune responses, difficult to induce with inactivated vaccines, can be effectively elicited with live vaccines, suggesting a strategy to enhance M2e immunogenicity in whole influenza virus vaccines. **Conclusions:** Finally, the successful development of the model rH5N2 vaccine strain is described; this strain provides dual protection, has potential applicability in regions where avian influenza is endemic, and can be used to promote the development of versatile H5N2 recombinant vaccines for effective avian influenza control.

## 1. Introduction

Highly pathogenic H5N1 avian influenza viruses (AIVs) have evolved into various neuraminidase (NA) subtypes (H5Nx), and the cocirculation of highly pathogenic (HP) H5Nx AIVs with low pathogenicity (LP) H9N2 AIVs is not uncommon [1,2]. Among these subtypes, the H5N1 and H9N2 subtypes are the most problematic, with reported cases of infections in various mammals, including humans [3,4,5]. In Korea, HP H5N1, H5N6, and H5N8 AIVs, and LP H9N2 AIV cause significant economic losses to the poultry industry, and vaccines against only H9N2 AI have been approved to date [6,7,8]. Recent studies have shown that while the hemagglutinin (HA) protein is the primary antigen in influenza vaccines, the neuraminidase (NA) protein also plays a crucial role in effective virus neutralization and protection [9,10]. However, vaccine development studies focusing on the immunogenicity of NA have primarily been conducted using platforms such as subunit vaccines, virus-like particles (VLPs), and recombinant viral vectors, which are composed of specific protein segments [11,12,13]. In contrast, studies investigating the immunogenicity of NA in whole inactivated vaccine platforms have been relatively limited. Moreover, the serological differentiation of infected from vaccinated animals (DIVA) remains an important prerequisite for developing vaccines against HP AIVs. Therefore, developing an inactivated H5N2 vaccine strain capable of effectively eliciting robust antibody responses against both HA and NA could play a crucial role in providing dual protection against H5Nx and H9N2 avian influenza viruses. Additionally, such a vaccine strain would be valuable for NA-based DIVA strategies for the detection of specific antibodies against N1, N6, and N8 [14].

Research has been performed to increase the viral titers and decrease the mammalian pathogenicity of vaccine strains. The balance between the HA and NA activities is crucial for efficient viral replication in embryonated chicken eggs (ECEs). N-glycan is acquired around the receptor-binding site (RBS) to help the virus evade humoral immunity by masking B-cell epitopes; however, this also decreases the binding affinity of HA to receptors (an α2,3-sialogalactose moiety) on the host cell surface [15]. To compensate for the decrease in the HA activity, the NA activity is often decreased by stalk deletion, resulting in reduced accessibility of NA to receptors [16]. Clade 2.3.2.1c A/mandarin duck/Korea/K10-483/2010 (K10-483) H5N1 viruses have acquired 144N-glycan and a 20-amino-acid deletion (aa-del) in the NA stalk [17]. An H9N2 vaccine strain, A/chicken/Korea/01310-CE20/2001 (01310), which belongs to a Y439-like lineage, was developed by passaging 20 times through ECEs. During passage, 01310 acquired an 18-aa-del in the NA stalk and a 133N-glycan in HA [18]. Moreover, the activity of polymerase protein complex of influenza A virus, composed of the trimer PB2, PB1, and PA, is primarily regulated by PB2 and strongly associated with the mammalian pathogenicity risk [19,20,21]. Our recent study revealed that the balance between polymerase activity and surface glycoprotein levels (HA and NA) is also critical for efficient viral replication [22]. Therefore, highly productive, nonmammalian pathogenic vaccine strains can be generated by optimizing the balance between the HA and NA activities and by ensuring proper alignment with polymerase activity [19].

The presence of universal epitopes across different subtypes of AIVs has been reported in the HA2 subunit and extracellular domain of the Matrix 2 (M2e) protein; here, they play a role in providing protection against the virus [23,24]. Although these universal epitopes exist, amino acid variations are present, and interestingly, N-glycans that can potentially mask these epitopes are commonly present in the HA (154N-glycan) of most viruses but are rarely found in M2e [25,26]. Notably, in the PR8 strain, which provides the six internal gene segments for recombinant vaccine strain development, M2e contains an N-glycan (20N-glycan, Appendix A). Therefore, the impact of N-glycan removal on the replication efficiency and cross-protection efficacy of vaccine strains need to be evaluated.

Different reagents are employed for the inactivation of vaccine viruses, and the choice of an appropriate inactivation method is crucial for vaccine efficacy. Formaldehyde (F/A) is a widely used inactivation reagent that works by covalently cross-linking the proteins and modifying the genetic material. However, the cross-linking of surface protective proteins may alter the antigenic structure and reduce the accessibility of epitopes, potentially affecting the antigenicity and immunogenicity of vaccines [27]. In contrast, binary ethylenimine (BEI) can inactivate viral genetic materials while preserving the function of viral proteins [28].

In this study, we aimed to develop a model vaccine strain capable of neutralizing both H5N1 and H9N2 viruses and suitable for serological differentiation of infected from vaccinated animals (DIVA) to differentiate major subtypes (H5N1, H5N6, and H5N8) of HP AIV infection. We hypothesized that modifying the HA2 stem, NA, and M2e regions could enhance vaccine immunogenicity. Specifically, the main strategies included removing the 154N-glycan from the HA2 stem, selecting an NA with no N-glycan in the stalk, introducing a chimeric PR8 M2 containing avian M2e, and deleting the 20N-glycan. Additionally, replacing the PB2 gene of the PR8 strain with that of the LP H9N2 strain was expected to reduce mammalian pathogenicity while increasing productivity in ECEs. Based on these assumptions, we developed the rvH5N2-aM2e-vPB2 vaccine strain and successfully generated a model recombinant H5N2 vaccine strain that demonstrated high productivity and nonpathogenicity in mammals. Additionally, compared with the F/A-inactivated vaccine, the BEI-inactivated rvH5N2-aM2e-vPB2 vaccine elicited significantly enhanced immune responses, particularly against NA; this resulted in superior virus neutralization activity against both H5N1 and H9N2. Finally, we successfully developed a model recombinant H5N2 vaccine strain that was highly productive, nonpathogenic in mammals, capable of neutralizing both H5N1 and H9N2, and suitable for application in the serological DIVA strategy.

## 2. Materials and Methods

### 2.1. Viruses, Cells and Plasmids

The A/chicken/Korea/01310-CE20/2001 (H9N2, 01310 E20) strain, passaged 20 times in specific pathogen-free (SPF) embryonated chicken eggs (ECEs), was obtained from the Laboratory of Influenza Viruses at the Animal and Plant Quarantine Agency (QIA) in Korea. This strain has been widely used as a vaccine to control outbreaks of low-pathogenicity H9N2 avian influenza viruses in Korea [18]. The high-pathogenicity strain A/mandarin duck/Korea/K10-483/2010 (K10-483), which was isolated from a migratory mandarin duck in 2010, is classified under clade 2.3.2.1c and caused a significant poultry outbreak in Korea from 2010 to 2011 [29]. Recombinant influenza viruses were generated using the Hoffmann vector system, passaged three times in 10-day-old SPF ECEs (VALO), and utilized for experiments [30]. The genomic segments used included the HA and NA genes of K10-483, the NA gene of 01310 E20, six internal genes (PB2, PB1, PA, NP, M, NS) from A/Puerto Rico/8/34 (H1N1, PR8), and the PB2 gene from 01310. All genomic segments were subsequently cloned and inserted into the pHW2000 vector system. The 293T and MDCK cell lines (obtained from the Korean Collection for Type Cultures (KCTC), Daejeon, Republic of Korea) were cultured in Dulbecco’s modified Eagle’s medium (DMEM, Gibco, Waltham, MA, USA) supplemented with 10% fetal bovine serum (FBS) and penicillin–streptomycin (Gibco). The cell lines were maintained at 37 °C under 5% CO_2_.

### 2.2. M2e Sequence Analysis

Sequences of the M2e region from H5N1, H5N6, H5N8, and H9N2 AIVs, which were isolated between 2018 and 2021, were obtained from the Global Initiative on Sharing All Influenza Data (GISAID). A total of 627 H5N1 M2e sequences, 364 H5N6 M2e sequences, 1453 H5N8 M2e sequences, and 993 H9N2 M2e sequences and the M2e sequence of PR8 were aligned for comparative analysis. The consensus sequences for each subtype, excluding the PR8 M2e, were determined. Potential N-glycosylation sites found in the consensus sequence were removed to determine the final M2e (Av) sequence.

### 2.3. Recombinant Virus Generation by Reverse Genetics

PR8-derived recombinant H5N1 viruses with different combinations of the attenuated HA gene of K10-483, the NA gene of K10-483 or 01310-CE20, the PB2 genes of PR8 or 01310, and the M2e (Av) gene and PR8 M2e gene were generated via reverse genetics. The HA gene of K10-483 was attenuated by replacing its polybasic cleavage site sequence (RERRRKR/GLF) with a monobasic sequence (ASGR/GLF). The M2e protein is highly conserved in AIVs and has become a major target for the development of universal vaccines [31,32,33]. However, we identified significant differences between PR8-M2e and avian-M2e, notably, a six-amino-acid discrepancy (Appendix A). Additionally, we discovered a potential N-glycosylation site at amino acid positions 20–22 in PR8 M2e, which could negatively impact vaccine immunogenicity. To incorporate a representative M2e (Av) sequence into rvH5N2-aM2e, we aligned the M2e sequence of the H5N1, H5N6, H5N8, and H9N2 subtype viruses isolated between 2018 and 2021, and these viruses pose major threats to the poultry industry; subsequently, we identified a consensus sequence (Appendix A). However, this consensus sequence contained potential N-glycosylation sites, which could shield epitopes and thereby adversely affect the immunogenicity of the vaccine [34]. Therefore, the potential N-glycan at the asparagine at position 18 was removed. Additionally, the N-glycosylation site covering the HA2 stem region was removed, especially for rvH5N2-aM2e. The genomic constellation of the recombinant viruses is summarized in Table 1. All recombinant viruses were generated using an 8-plasmid reverse genetics system [30]. The 293T cells were cultured at a density of 1 × 10^6^ cells per well in 6-well plates and transfected with 300 ng of each plasmid using Lipofectamine 2000 (Invitrogen, Waltham, MA, USA) and PLUS Reagent (Invitrogen) in 1 mL of Opti-MEM (Gibco). After 24 h of incubation, the cells were supplemented with fresh Opti-MEM containing L-1-tosylamido-2-phenylethyl chloromethyl ketone-treated trypsin (TPCK-treated trypsin, Sigma-Aldrich, St. Louis, MO, USA). Following an additional 24 h incubation, 200 µL of the culture supernatant was inoculated into 10-day-old SPF ECEs and was then incubated at 37 °C for three days. After incubation, allantoic fluid was collected and subjected to a hemagglutination assay using 1% chicken red blood cells (RBCs), following the protocols outlined in the WHO Manual on Animal Influenza Diagnosis and Surveillance. Each recombinant virus generated was verified using real-time polymerase chain reaction (RT-PCR) and Sanger sequencing as previously reported [35].

### 2.4. Recombinant Virus Titration in ECEs

Each recombinant virus was serially diluted 10-fold using PBS, starting from a 10^6^-fold dilution to a 10^9^-fold dilution, and each dilution was inoculated into five 10-day-old SPF ECEs to measure the titer of the recombinant viruses. The presence of AIVs in the allantoic fluid was confirmed by the HA assay. The 50% infectious dose (EID_50_/mL) in chicken embryos was calculated via the Spearman–Karber method [36]

### 2.5. Growth Curves of the Recombinant Viruses in the MDCK Cells

To measure the infectivity of the recombinant viruses in mammalian cell lines, each virus was inoculated into the MDCK cells in a 12-well plate at an MOI of 0.001. After incubation for one hour at 37 °C in a 5% CO_2_ incubator, the inoculum was replaced with fresh medium. The supernatants were then collected at intervals of 0, 24, 48, and 72 h. These collected supernatants were subjected to 10-fold dilution and subsequently inoculated into MDCK cells to determine the viral titer, which was measured as the TCID_50_/mL.

### 2.6. Virus Inactivation Using Formaldehyde or BEI

Each recombinant virus was used in undiluted allantoic fluid, with the following viral titers used for the viruses: rH5N1-vPB2 = 10^9.33±0.23^ EID_50_/mL, rH5N2-vPB2 = 10^9.0±0.35^ EID_50_/mL, and rvH5N2-aM2e-vPB2 = 10^9.58±0.23^ EID_50_/mL (Table 1). The recombinant viruses from each group were then inactivated via two different methods. For formaldehyde-mediated inactivation, 0.2% formaldehyde was applied to the viruses, and these viruses were then incubated in a 37 °C incubator overnight. Viral inactivation was confirmed by inoculating the sample into SPF ECEs. For BEI-mediated inactivation, 0.1 M BEI was mixed with the viruses, followed by a 24 h incubation in a 37 °C incubator. Inactivation was terminated by the addition of a 1 M sodium thiosulfate solution. Viral inactivation was confirmed using SPF ECEs; this process was consistent with previously described procedures. The two types of vaccines were prepared by mixing with ISA78 (SEPPIC) at a 3:7 ratio to create oil emulsion vaccines.

### 2.7. Immunogenicity of the H5N2 Recombinant Viruses

To evaluate the immunogenicity of formaldehyde- or BEI-inactivated H5N2 recombinant viruses, five three-week-old SPF chickens were vaccinated via intramuscular (IM) injection with 0.5 mL of inactivated virus. Blood samples were collected from the wing vein at 0, 7, 14, 21, and 28 days post-vaccination (DPV). Hemagglutination inhibition (HI) tests were performed according to the WHO Manual on Animal Influenza Diagnosis and Surveillance. Briefly, each serum sample was treated at 56 °C for 30 min and diluted 2-fold with phosphate buffered saline (PBS), and 25 μL of each diluted sample was mixed with an equal volume of 4 hemagglutinating units (HAUs) of virus. After incubation at room temperature for 30 min, 25 μL of 1% (*v*/*v*) chicken red blood cells (RBCs) was added, and hemagglutination was recorded after 40 min.

To assess the M2e immune response and protective efficacy of live H5N2 recombinant virus inoculation, six-week-old female BALB/c mice (*n* = 13) were intranasally administered either 10^4^ EID_50_ of live virus (rH5N2 or rvH5N2-aM2e) or PBS as a negative control. Two weeks after virus inoculation, sera were collected from five mice to evaluate the M2e immune response, and the remaining mice (*n* = 8) were challenged with 10^6^ EID50 of SNU50-5 (A/wild duck/Korea/SNU50-5/2009(H5N1)), PR8 (A/Puerto Rico/8/1934(H1N1)), or PR8-M(Av) (PR8 virus with avian M2e). Body weight and survival rates were monitored for an additional two weeks. At three days post-challenge (dpc), three mice were sacrificed, and the lung viral titers were determined via TCID_50_/mL measurements.

### 2.8. Virus Neutralization (VN) Test

A virus neutralization (VN) assay was conducted via a previously described method with slight modifications [37]. Heat-inactivated serum was diluted two-fold in DMEM, without the addition of FBS, in a 96-well U-plate (starting with a 32-fold dilution in the first well). The diluted serum was mixed with an equal volume of 100 TCID_50_ antigen and incubated at 37 °C for 1 h. The mixture was subsequently added to a 96-well plate containing a monolayer of MDCK cells and incubated at 37 °C for 3 days. On the third day, the supernatant was harvested and mixed at a 1:1 ratio with 1% chicken RBCs. The VN titer was defined as the highest serum dilution that inhibited the hemagglutination of the RBCs.

### 2.9. M2e-Specific IgG Analysis Using Enzyme-Linked Immunosorbent Assay (ELISA)

To assess M2e-specific IgG titers, an M2e peptide-coating ELISA was performed. Peptides with sequences corresponding to PR8 M2e (MSLLTEVETPIRNEWGCRCNGSSD) and M2e (Av) (MSLLTEVETP**T**RN**G**W**E**C**K**C**SD**SSD) were synthesized (BIONICS, Seoul, Republic of Korea). Both M2e peptides were coated onto a 96-well immunoplate and incubated overnight at 4 °C. On the next day, the plate was washed with PBS containing 0.1% Tween 20 (PBST). The plate was subsequently blocked with blocking buffer (PBST with 0.1% BSA) and incubated at room temperature for 2 h. Serum samples were subjected to two-fold serial dilutions in fresh 96-well U-plates. These diluted serum samples were then transferred to a blocking plate and incubated at room temperature for 1 h. After the plates were washed, the horseradish peroxidase (HRP)-conjugated secondary antibody was applied to all wells, and the plate was incubated for an additional hour at room temperature. Subsequently, the HRP reaction was induced using the substrate TMB. The reaction was stopped with 0.1 M H_2_SO_4_, and the absorbance at 450 nm was measured with a microplate reader (TECAN, Zürich, Switzerland). To evaluate the antibody responses to M2e following live virus inoculation, sera collected from mice at two weeks post-inoculation were analyzed via M2e peptide-coated ELISA following the same procedure as that used for chicken sera.

### 2.10. NA Inhibition Test

The NA inhibition activity of the serum samples was measured using the NA-star™ Influenza Neuraminidase Inhibitor Resistance Detection Kit (Thermo Fisher Scientific, Waltham, MA, USA) according to the manufacturer’s instructions. Briefly, heat-treated serum samples collected at 3 weeks post vaccination (wpv) were subjected to two-fold serial dilutions in NA star assay buffer, starting from a 1:16 dilution to a 1:1024 dilution. Viruses were added to the serially diluted serum samples, and these samples were subsequently incubated at room temperature for 20 min. Then, 1:1000 diluted NA substrate was added to all wells at 10 μL/well, and the plate was incubated at room temperature for 30 min. Then, the NA-star accelerator was added at 60 μL/well, and luminescence was measured immediately using an Infinite 200 PRO (TECAN, Switzerland). The serum dilution titer that achieved 50% inhibition of NA activity was 50% inhibitory concentration (IC_50_) against the virus.

### 2.11. T-Cell Epitope Analysis and Structural Modeling

To compare the CD8^+^ T-cell epitopes of the HA, NA, NP, M1, and NEP proteins between the vaccine strains (rH5N2/rvH5N2-aM2e) and challenge viruses (SNU50-5, PR8) used in the mouse study, the Immune Epitope Database Analysis Resource (http://tools.iedb.org/main/tcell/ (accessed on 19 September 2024)) was utilized, with the allele set to H-2-Kd. Predicted epitopes with a percentile rank below 0.5 were considered to possess high MHC affinity.

The 3D structures of the N1 protein from the rH5N1 virus and the N2 protein from the 01310 (H9N2) virus were predicted using the AlphaFold 3 model. The predicted structures were visualized in PyMOL v4.6.0. To compare the heights of the ectodomains (head and stalk) of the NA, excluding the transmembrane and cytoplasmic tail regions, the average distance between five conserved active-site amino acids (R118, R152, E276, R292, and R371) in the head and the amino acid at the lowest position in the stalk was measured.

### 2.12. Ethics Statement

The experiments involving SPF chickens and BALB/c mice were conducted in accordance with the guidelines approved by the Institutional Animal Care and Use Committee (IACUC) of Seoul National University (IACUC-SNU-230612-5, SNU-220412-1-2) and were designed to minimize animal suffering.

SPF chicken vaccination experiments were carried out in an animal biosafety level 1 facility at the College of Veterinary Medicine, Seoul National University (Seoul, Republic of Korea). BALB/c mouse live vaccine administration and challenge studies were conducted in an animal biosafety level 2 facility at the Animal Center for Pharmaceutical Research, Seoul National University (Seoul, Republic of Korea). To ensure animal welfare, the chickens and mice were monitored daily, and water and feed were provided in sufficient quantities and replenished at least once a week.

### 2.13. Statistical Analysis

Statistical plots were generated via GraphPad Prism 9.5.1 and Q-Q plots were used to check for normal distribution. All data are presented as the means ± SDs. The replication efficacy of the recombinant viruses in embryonated eggs, NA inhibition test results, and mouse lung viral titers were compared between the groups via one-way ANOVA. For comparing growth kinetics in the MDCK cell line and for the HI and VN assays and ELISAs, two-way ANOVA was employed.

## 3. Results

### 3.1. Generation and Comparison of the Replication Efficiency of Recombinant Viruses in ECEs to Develop a Dual-Protective Vaccine Against Different AIV Subtypes

We generated a clade 2.3.2.1c H5N1 recombinant strain with HA and NA genomic sequences from K10-483, six internal genes from PR8 (rH5N1), and a variant with the 01310 PB2 gene, and the PR8 PB2 gene (rH5N1-vPB2) was replaced. As previously reported, the viral titer of rH5N1-vPB2 was approximately 12-fold greater (10^9.33±0.23^ EID_50_/mL) than that of rH5N1 (10^8.25±0.47^ EID_50_/mL) [17]. Additionally, four recombinant H5N2 vaccine strains possessing 01310 NA2 were successfully generated (Table 1). The 01310 NA2 sequence with an 18-aa deletion and the absence of N-glycan in the stalk appeared to be compatible with clade 2.3.2.1c HA and the six internal genes of PR8; this resulted in a significantly higher viral titer for rH5N2 (10^9.75±0.40^ EID_50_/mL) than for rH5N1. However, replacing PR8 PB2 with 01310 PB2 decreased the viral titer of rH5N2-vPB2 with respect to that of rH5N2. The viral titer of rH5N2-vPB2 was not significantly greater than that of rH5N1. Additionally, the rvH5N2-aM2e and rvH5N2-aM2e-vPB2 strains lacked the 154N-glycan in the HA2 subunit and featured a chimeric PR8 M2 containing the avian M2e epitope (M2e (Av)) (Appendix A); these strains had significantly higher viral titers than rH5N1, by approximately 38-fold (10^9.83±0.31^ EID_50_/mL) and 23-fold (10^9.58±0.23^ EID_50_/mL), respectively. Notably, M2e (Av) increased the viral titer of rvH5N2-aM2e-vPB2 with respect to that of rH5N2-vPB2.

### 3.2. Inhibition of the Replication of the Recombinant Vaccine Strains in the MDCK Cells Using the 01310 PB2 Gene

To assess the potential mammalian infection risk of the developed recombinant vaccine strains, the replication efficiencies of the recombinant viruses were compared in a mammalian cell line, namely, MDCK cells. None of the recombinant viruses containing the 01310 PB2 gene (rH5N1-vPB2, rH5N2-vPB2, and rvH5N2-aM2e-vPB2,) grew at all over 72 h. All strains with the PR8 PB2 gene (rH5N1, rH5N2, and rvH5N2-aM2e) grew in the MDCK cells. Notably, compared to that of rH5N1, the replication efficiency of rH5N2 and rvH5N2-aM2e was significantly greater at all time points, and the replication efficiency of rvH5N2-aM2e reached nearly 10^8^ TCID_50_/mL at 48 hpi (Figure 1). These findings indicated that rH5N2-vPB2 and rvH5N2-aM2e-vPB2 posed minimal risks of mammalian infection, showing their potential suitability as vaccine strains.

### 3.3. Comparison of the HI Antibody Titers Induced by Different Vaccine Strains and Inactivation Methods

rH5N1-vPB2, rH5N2-vPB2, and rvH5N2-aM2e-vPB2 were inactivated with either formaldehyde or BEI, and oil emulsion vaccines were prepared. These vaccines were used to inoculate three-week-old SPF chickens (n = 5), and the serum samples were collected at 1–4 weeks postvaccination (wpv). HI antibody titers were measured against the rH5N1 and 01310 (H9N2) strains (Figure 2). The HI antibody titers against rH5N1 were negligible at 1 wpv but sharply increased to approximately 256 or higher from 2 wpv across all vaccine groups regardless of the inactivation method used (Figure 2A). The rH5N2-vPB2 vaccine group, which exhibited lower viral titers than other groups, showed the lowest HI antibody titers against rH5N1 among all vaccine groups at 3 and 4 wpv, regardless of the inactivation method. No significant difference was observed in the HI antibody titers between the two different inactivation methods for the same vaccine strain. The HI antibody titers for 01310 (H9N2) were uniformly undetectable across all groups, with the exception of the BEI-inactivated rH5N2-vPB2 and rvH5N2-aM2e-vPB2 vaccine groups; these groups had very low but positive HI antibody titers ranging from 2 to 6 (Figure 2B). Overall, regardless of the inactivation method or the type of vaccine strain, high HI titers were observed against H5N1, whereas almost no detectable HI titers were observed against H9N2.

### 3.4. Comparison of the VN Antibody Titers Induced by Different Vaccine Strains and Inactivation Methods

The VN test was conducted with the same serum samples and the virus antigens used for the HI test (Figure 3). The VN antibody titers against rH5N1 were already greater than 2^8^ (256) at 2 wpv and gradually increased to greater than 2^11^ (2048) at 4 wpv in all vaccine groups (Figure 3A). No significant differences were observed in the VN antibody titers between the two different inactivation methods for the same vaccine strain. Similar to the HI test results, the rH5N2-vPB2 group showed the lowest VN antibody titers against rH5N1 at 3 and 4 wpv compared to all other vaccine groups. Notably, at 3 wpv, the rH5N2-vPB2 group, inactivated with either F/A or BEI, showed significantly lower VN antibody titers compared to the BEI-inactivated rH5N1-vPB2 vaccine group. At 4 wpv, VN antibody titers of the rH5N2-vPB2 group, inactivated with either F/A or BEI, were significantly lower than those of the F/A-inactivated rH5N1-vPB2 and BEI-inactivated rH5N1-vPB2 groups. No significant difference was observed in the VN titers between the rH5N1-vPB2 and rvH5N2-aM2e-vPB2 vaccine groups. In the VN test against 01310 (H9N2), the rH5N1-vPB2 vaccine group, regardless of inactivation, consistently showed negligible VN antibody responses (Figure 3B). Notably, the BEI-inactivated rvH5N2-aM2e-vPB2 vaccine group had significantly higher VN antibody titers than the other vaccine groups. The rvH5N2-aM2e-vPB2 vaccine group had VN antibody titers that were comparable to those of the rH5N2-vPB2 group against the 01310 (H9N2) antigen when inactivated with F/A; however, inactivation with BEI resulted in a significant increase in the VN antibody titers, to even greater than 2^8^ (256) at 3 and 4 wpv.

### 3.5. Improved NA Immunogenicity of the BEI-Inactivated rvH5N2-aM2e-vPB2 Vaccine

To assess whether the increased VN antibody titers against 01310 (H9N2) observed in the BEI-inactivated rvH5N2-aM2e-vPB2 vaccine group were caused by increased immunogenicity of NA, NI tests were used to measure anti-N1 (rH5N1) and anti-N2 [01310 (H9N2)] antibodies that inhibited NA activity (Figure 4). According to the N1 inhibition test results, the BEI-inactivated rH5N1-vPB2 vaccine group consistently had greater NI activity across various serum dilutions than the other vaccine groups (Figure 4A). A comparison of the IC_50_ values revealed that the serum concentration required for 50% inhibition of NA activity was significantly greater in the BEI-inactivated rH5N1-vPB2 group compared to groups vaccinated with other strain (Figure 4C). In the N2 inhibition test, the rH5N2-vPB2 and rvH5N2-aM2e-vPB2 vaccine groups showed greater NI activity than the rH5N1-vPB2 vaccine group regardless of the inactivation method in all the serum dilution ranges. Notably, the rvH5N2-aM2e-vPB2 vaccine induced the most potent NI activity across all groups, and this effect was more pronounced when the vaccine was inactivated with BEI rather than F/A (Figure 4B). The IC_50_ value for the BEI-inactivated rvH5N2-aM2e-vPB2 vaccine group was significantly greater than that for all the other vaccine groups (Figure 4D). These findings indicated that rvH5N2-aM2e-vPB2, particularly when inactivated with BEI, prompted the production of a robust N2-inhibiting antibody, which could contribute to 01310 (H9N2) virus neutralization efficacy to some extent. Interestingly, the NI antibody titer of the BEI-inactivated rH5N1-vPB2 vaccine against N1 (4.7 log2) was lower than that of the BEI-inactivated rvH5N2-aM2e-vPB2 vaccine against N2 (6.6 log2), and the immunogenicity of N1 was likely lower than that of N2.

### 3.6. Detectable Anti-M2e Antibody Induction by Live rH5N2-vPB2 and rvH5N2-aM2e-vPB2 Infections

To assess whether the increase in VN antibody titers against 01310 (H9N2) observed in the rvH5N2-aM2e-vPB2 vaccine group was caused by the anti-avian M2e antibodies, which were induced by increased immunogenicity and antigenic homogeneity, a peptide ELISA coating of two different epitopes from PR8 and avian M2e was conducted (Figure 5). Compared with those of the control group, the OD values of the inactivated rH5N2-vPB2 and rvH5N2-aM2e-vPB2 vaccine groups were not significantly different according to both the PR8 and avian M2e peptide ELISAs (Figure 5A–D). These findings indicated that the increased VN titers in the rvH5N2-aM2e-vPB2 vaccine group were not likely caused by the anti-avian M2e antibodies. Additionally, inactivated whole-virus vaccines clearly could not induce the production of detectable anti-M2e antibodies.

To compare the response to M2e without virus inactivation, we infected BALB/c mice with live rH5N2 or rvH5N2-aM2e. The serum samples were collected at 2 weeks post inoculation (wpi) to assess anti-M2e antibody responses (Figure 5E,F). Since the rH5N2-vPB2 and rvH5N2-aM2e-vPB2 strains were predicted to inefficiently replicate in BALB/c mice due to the 01310 PB2 gene, they were replaced with rH5N2 and rvH5N2-aM2e (Figure 1) [38]. The serum samples collected from the rH5N2- and rvH5N2-aM2e-infected groups had significantly higher anti-M2e antibody levels than those from the control group according to both the PR8 and avian M2e peptide ELISAs. Interestingly, the anti-avian M2e antibody level was significantly greater in the rvH5N2-aM2e infection group than in the rH5N2 infection group at a serum dilution of 2^6^ (Figure 5F).

### 3.7. Comparison of Live rH5N2 and rvH5N2-aM2e Vaccine Efficacies in BALB/c Mouse Model

Although rH5N2 and rvH5N2-aM2e were developed as inactivated vaccines for poultry, we evaluated their efficacy as live vaccines in a mouse model to investigate the impact of the introduced mutations on heterosubtypic protection. We inoculated live rH5N2 and rvH5N2-aM2e to BALB/c mice intranasally, and the mice were challenged with the H1N1 strains PR8 and PR8-M(Av) and an LP H5N1 strain (A/wild duck/Korea/SNU50-5/2009, SNU50-5) at 2 weeks post-inoculation (wpi). The PR8-M(Av) strain is a chimeric recombinant PR8 strain possessing a mutated PR8 M (aM2e). The SNU50-5 strain is a wild-type H5N1 virus that contains the avian M2e sequence. In contrast to the negative control, all live vaccines protected the mice from body weight loss and mortality and significantly decreased viral replication in the lungs after the mice were lethally challenged with the H1N1 and H5N1 strains (Figure 6). Interestingly, compared with that in the rH5N2 vaccine group, the viral titer in the rvH5N2-aM2e vaccine group was significantly lower when the animals were challenged with SNU50-5 (Figure 6I). The amino acid identities of SNU50-5 HA with the vaccine strain HA were 86% (HA1) and 94.5% (HA2). Since live vaccines reportedly induced cellular immunity in addition to the humoral immune response [39], we performed a comparative analysis of CD8^+^ T-cell epitopes in the major proteins of the challenge and vaccine strains (Table 2 and Appendix A). The numbers of CD8^+^ T-cell epitopes of PR8 and SNU50-5 identical to those of the vaccine strains were 4 (HA), 6 (NP), 4 (M1), and 3 (NEP), and 9 (HA), 4 (NP), 3 (M1), and 1 (NEP), and the total number for each were the same at 17 (Table 2).

## 4. Discussion

Clade 2.3.2.1c H5N1 viruses were evolved from clade 2.3.2 and represent the accumulation of adaptive mutations for over a decade of repeated infections in chicken flocks [40]. These viruses caused fatal human cases in Cambodia in 2024 and have become endemic in several Asian countries [41]. Furthermore, progeny variants of clade 2.3.2.1c, such as clades 2.3.2.1d and 2.3.2.1e, have emerged in China [42]. As previously reported, the viral titer in ECEs of the clade 2.3.2.1c H5N1 vaccine strain generated through conventional reverse genetics using the six internal genes of PR8 was very low. However, the viral titer could be increased more than ten-fold by replacing the PR8 PB2 gene with the 01310 PB2 gene [17]. Therefore, the continuous circulation of clade 2.3.2.1c and its progeny variants reflects the need for high-titer vaccines as well as massive, nationwide vaccination.

The unexpectedly high viral titers of rH5N2 and rvH5N2-aM2e compared to rH5N1 in ECEs support the importance of the balance of the HA and NA activities with each other as well as with the activity of PB2 (Table 1). The increase in viral titer may be attributed to differences in the enzyme activities of N1 and N2, which may cooperate with HA to affect viral replication efficiency. Most clade 2.3.2.1c H5N1 viruses also acquired the V223I mutation at the interfaces of HA trimer globular heads, and as a result, the thermostability of HA decreased [40]. The thermostability of HA was related to acid stability, and clade 2.3.2.1c HA was predicted to change conformation for fusion at relatively high pH [43]. This characteristic suggests that the optimal balance observed with H5 paired with N2, rather than N1, might not solely result from the balance between HA and NA activities. The role of the activation pH of HA and NA may be an influential factor in viral fitness. For the fusion of the viral envelope and the endosomal membrane, HA needed to detach from the receptor to induce destabilization of the globular head trimer. Previous studies have demonstrated that the optimal pH for the enzymatic activity of most N1s is slightly lower than that of N2. A shorter N1 may be inactive at the pH of the early endosome, potentially hindering HA release from receptors [44]. However, N2, which exhibits optimal sialidase activity at relatively higher pH levels, is likely more effective than N1 in releasing receptors before clade 2.3.2.1c HA undergoes conformational changes. The role of various types of HA in influencing viral envelope and endosomal membrane fusion as endolysosomal pH decreases has been well studied [45]. However, investigating how different types of NAs might influence the viral envelope and endosomal membrane fusion as the endolysosomal pH decreases would be particularly interesting. Although we did not directly compare the enzyme activities, the length of the NA stalk is one of the factors that influences receptor accessibility and neuraminidase activity [46]. Both N1 of K10-483 and N2 of 01310 underwent adaptation in poultry flocks, resulting in 20- and 18-amino-acid deletions in their stalks, respectively. When the lengths of the extracellular regions of these NA proteins were compared, N1 (56.4 Å) was shorter than N2 (64.0 Å) (Appendix A). Stalk amino acid deletions could reduce the NA activity, and shorter NAs could have a lower probability of cleaving host cell surface receptors for rapid viral escape. The reduced activity of N1 could lead to an imbalance in the stronger activities of HA and PB2, resulting in decreased replication efficiency of rH5N1. In this context, compared with that of PR8 PB2, the relatively low viral titer of rH5N2-vPB2 could be attributed to the relatively low activity of 01310 PB2, leading to an imbalance [20]. The effect of lower PB2 activity could potentially be balanced by the combined mutations in HA (TGT) and M2e (aM2e), which increase the viral replication efficiency of rvH5N2-aM2e-vPB2. The TGT mutation at the 154N-glycosylation site is a rare mutation that affects virus replication efficiency to a lesser extent than many other mutations in our previous study [47]. Previous studies have demonstrated that the M2 protein significantly influences the viral budding capacity and contributes to viral assembly through M1 protein recruitment to the cell membrane [48,49]. To date, experimental evidence supporting the effect of M2e modification on viral replication efficiency has not been reported; however, compared with that of PR8 (mean death time, 129.6 h), the delayed mortality of PR8-M(Av) (mean death time, 153.6 h) potentially indicated that M2e modification could decrease viral fitness (Figure 6).

The PR8 PB2 gene contains multiple mammalian pathogenicity-related mutations, including the most potent E627K mutation, and intentional or accidental generation of artificial reassortants of the AIV strains possessing PR8 PB2 need to be avoided [20]. In contrast to the PR8 PB2-possessing recombinant strains, all 01310 PB2-possessing recombinant strains did not replicate in the MDCK cells. The effect of N2 on viral replication efficiency was also apparent in the MDCK cells, and rH5N2 and rvH5N2-aM2e had significantly higher viral titers than rH5N1 (Figure 1). Notably, rH5N1 replicated in the lungs of BALB/c mice without body weight loss in our previous study, and rH5N2 and rvH5N2-aM2e did not cause body weight loss after inoculation at 10^4^ EID_50_/mouse in this study [17]. However, all G1-, Y280-, and Y439-like H9N2 recombinant vaccine strains possessing PR8 PB2 caused severe body weight loss in BALB/c mice [22,50,51]. Therefore, the biosafety of vaccine strains and the biosecurity of vaccine production lines for veterinary use need to be ensured, especially when they represent nonvaccine subtypes in humans.

The immunogenicities of rH5N1-vPB2, rH5N2-vPB2, and rvH5N2-aM2e-vPB2 were sufficiently high to induce very high HI and VN titers against rH5N1 irrespective of the inactivation reagents used; however, BEI-inactivated rvH5N2-aM2e-vPB2 was optimal because it induced significantly higher VN and NI antibody levels against 01310 (H9N2) than the other vaccines (Figure 3 and Figure 4). Formaldehyde, with its propensity to induce cross-linking between proteins, likely altered viral antigenic epitopes to potentially impacting the vaccine antigenicity and efficacy [27,28]. Similar to formaldehyde, BEI, which is primarily used as an inactivation reagent for foot-and-mouth disease virus, is effective at inactivating most viruses within 24 h at 37 °C, even at low concentrations. However, unlike formaldehyde, BEI does not induce cross-linking between the proteins when it is used at low concentrations. Instead, BEI inactivates viruses by alkylating their internal genetic material, thereby preserving viral antigenicity with minimal impact [28,52,53]. These properties contributed to the robust immune response against neuraminidase (NA) observed with BEI-inactivated rvH5N2-aM2e-vPB2. The enhanced NA-specific immune response likely led to the induction of increased neutralizing antibody titers against 01310 (H9N2). Additionally, in this study, we aimed to evaluate the impact of N-glycan removal from the HA, NA stalk, and M2e of rvH5N2-aM2e-vPB2 on enhancing immunogenicity. While some effects of N-glycan removal were observed in our results, the difference in immunogenicity could not be fully explained by these effects alone. The differences in immunogenicity between rH5N2-vPB2 and rvH5N2-aM2e-vPB2 for the homologous antigen (rH5N1) could not be attributed to the effects of N-glycan removal due to differences in the viral titers administered. Similarly, the differences in immunogenicity between rH5N1-vPB2 and rvH5N2-aM2e-vPB2 against rH5N1 could not be explained by the effects of N-glycan removal, likely due to the overall high levels of immunogenicity of both vaccines. On the other hand, for each homologous NA antigen, NI antibody level of the BEI-inactivated rH5N1-vPB2 vaccine was lower than of the BEI-inactivated rvH5N2-aM2e-vPB2 vaccine; this result supported previous findings that a shorter NA stalk is associated with reduced NA immunogenicity (Appendix A) [54]. Moreover, higher NI antibody levels against homologous NA subtypes could be useful in the serological DIVA strategy to diagnose unvaccinated H5N1 HPAIV infection (Figure 4C,D) [55,56].

Anti-M2e antibody production was not induced by inactivated oil emulsion vaccines but was instead induced by live vaccines. The M2 protein is abundantly expressed on the surface of infected cells but is only sparsely incorporated into virions [57]. Therefore, the lack of immunogenicity of M2e on virions was attributed to its low copy number and limited surface exposure; this phenomenon could be overcome by using live vaccines, as demonstrated in this study. This finding could also be useful for verifying the M2e-based serological DIVA strategy. The M2e epitopes recognized by monoclonal antibodies are composed of four amino acid residues: E6, P10, I11, and W15. In this epitope, only one amino acid of PR8 M2e (T11) differed from that of M2e (Av) (I11) [58]. However, M2e(Av) contained additional mutations, including E14G, G16E, R18K, N20S, and G21D; this result differed from the residues in PR8 M2e. Despite these multiple amino acid differences, the mouse anti-serum samples collected from the live rH5N2- and rvH5N2-aM2e-vaccinated groups had similar OD values in the PR8 M2e peptide ELISA (Figure 5E); these results indicated that mutations other than I11T were unlikely to affect the antibody binding. However, the anti-serum samples from the rvH5N2-aM2e-vaccinated group had significantly greater OD values than those from the rH5N2 group according to the M2e(Av) peptide ELISA (Figure 5F). The live rvH5N2-aM2e vaccine could exhibit enhanced immunogenicity caused by the reduction in the N-glycan levels in HA, NA, and M2e; this reduction subsequently increased the binding affinity and facilitated the generation of more specific antibodies against the M2e(Av) peptide within just 14 days. Furthermore, the higher M2e antibody titers induced by rvH5N2-aM2e could have compensated for the lower binding affinity to the PR8 M2e peptide, resulting in OD values comparable to those of rH5N2 against PR8 M2e. The live rH5N2 and rvH5N2-M2e vaccines provided complete protection against the heterosubtypic PR8 and PR8-M(Av) strains, which possessed nearly identical internal genes; additionally, the live rH5N2 and rvH5N2-M2e vaccines provided complete protection against the SNU50-5 strain, which had different internal genes. The protective efficacy of live vaccines could be attributed mainly to common CD8^+^ T-cell epitopes related to cellular immunity (Table 2 and Appendix A). This aligns with previous studies highlighting the critical role of cellular immunity in protection and the advantage of live vaccines in inducing robust CD8^+^ T-cell responses [39,59]. However, the significantly lower lung viral titer in the rvH5N2-aM2e-vaccinated group than in the rH5N2 group could not be explained solely by cellular immunity since they shared identical CD8^+^ T-cell epitopes. Additionally, while antibodies against M2e lacked neutralizing activity compared with antibodies against HA, they could bind to M2e expressed on virus-infected host cells and inhibit viral budding through antibody-dependent cellular cytotoxicity (ADCC), thereby reducing viral replication [60,61]. Therefore, the increased levels of anti-M2e(Av) antibodies could contribute to the reduced lung viral titers of both PR8-M(Av) and SNU50-5 (Figure 6H,I). Overall, we demonstrated an effective strategy for inducing M2e-specific immune responses through live vaccine study. However, the live vaccines used in this study are recombinant viruses, which pose a significant biosafety concern due to the potential risk of reassortment. To address this issue, it is essential to develop recombinant live vaccines with genetic modifications that minimize the likelihood of reassortment, ensuring their safety for practical application.

In our study, we did not assess the protective efficacies of developed vaccines in chicken challenge studies, and further study is required to conclude the vaccine’s efficacy. If the H5N2 vaccine’s efficacy is verified in the animal experiment, our results may promote the development of other H5N2 vaccine strains covering clade 2.3.4.4b H5Nx and Y280-like H9N2 or G1-like H9N2 vaccine strains. In addition, this dual-protective H5N2 vaccine strain could reduce the antigen volume needed for avian influenza vaccines, creating space to include additional vaccines in multivalent oil emulsion vaccines for poultry against Newcastle disease, infectious bronchitis, egg drop syndrome, metapneumovirus infection, and avian influenza.

## 5. Conclusions

We succeeded in generating a highly productive mammalian nonpathogenic H5N2 vaccine strain providing dual protection against H5Nx and H9N2 AIVs. Additionally, we demonstrated that determining the proper inactivation reagent is critical for optimizing the immunogenicity of whole inactivated vaccines.

## Figures and Tables

**Figure 1 vaccines-13-00022-f001:**
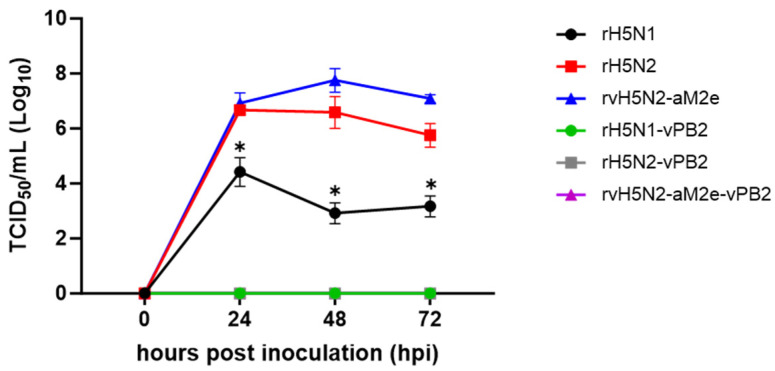
Comparison of the replication efficiencies of the recombinant viruses in MDCK cells. MDCK cells were inoculated with recombinant H5N1 or H5N2 virus at an MOI of 0.001. After 1 h of incubation, the inoculum was replaced with fresh medium, and the supernatant was obtained at each time point (0, 24, 48, and 72 h). The viral titer was measured as the TCID_50_/mL in the MDCK cells, and the results are presented as the means ± SDs of triplicate experiments. Statistical significance was analyzed by two-way ANOVA. The asterisk represents a significant difference between rH5N1 and the other groups (*p* < 0.001).

**Figure 2 vaccines-13-00022-f002:**
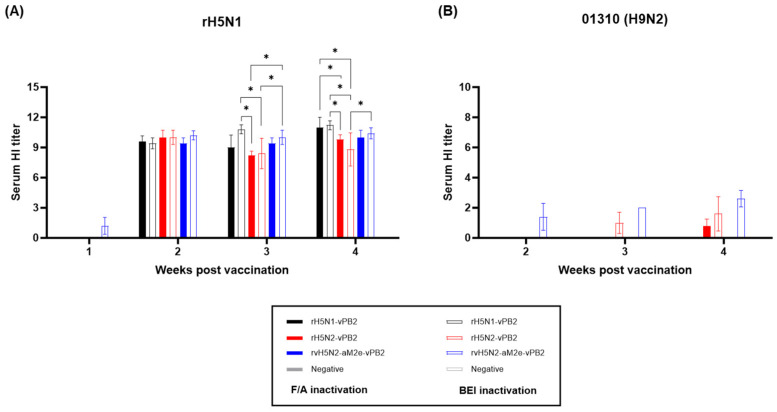
Comparison of the serum HI titers induced by recombinant virus vaccines inactivated by formaldehyde (F/A) or binary ethylenimine (BEI). Comparison of the HI titers at 1–4 weeks postvaccination. The serum samples were collected from SPF chickens (n = 5). (**A**) HI antibody responses against rH5N1 induced by vaccines inactivated with F/A or BEI. (**B**) HI antibody responses against 01310 (H9N2) induced by vaccines inactivated with F/A or BEI. The HI antibody response induced by the F/A-inactivated vaccine is represented by solid bars, while the response induced by the BEI-inactivated vaccine is represented by open bars. The data are presented as the means ± SD. Statistical significance was analyzed by two-way ANOVA and is denoted by asterisks (* *p* <0.01).

**Figure 3 vaccines-13-00022-f003:**
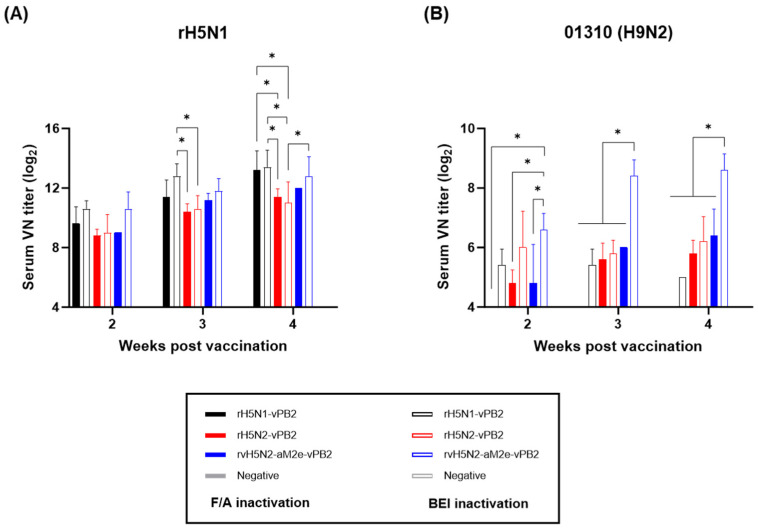
Comparison of the serum VN titers induced by recombinant virus vaccines inactivated by formaldehyde (F/A) or binary ethylenimine (BEI). Serum samples (n = 5) collected at 2, 3, and 4 weeks post-vaccination were utilized to conduct virus neutralization (VN) tests. (**A**) VN antibody responses against rH5N1 induced by vaccines inactivated with either F/A or BEI. (**B**) VN antibody responses against 01310 (H9N2) induced by vaccines inactivated with either F/A or BEI. We distinguished between the two vaccine groups in the bar graphs; the vaccines inactivated with F/A are represented by solid bars, whereas those inactivated with BEI are depicted by open bars. The data are presented as the means ± SD. Statistical significance was analyzed by two-way ANOVA and is denoted by asterisks (* *p* < 0.01).

**Figure 4 vaccines-13-00022-f004:**
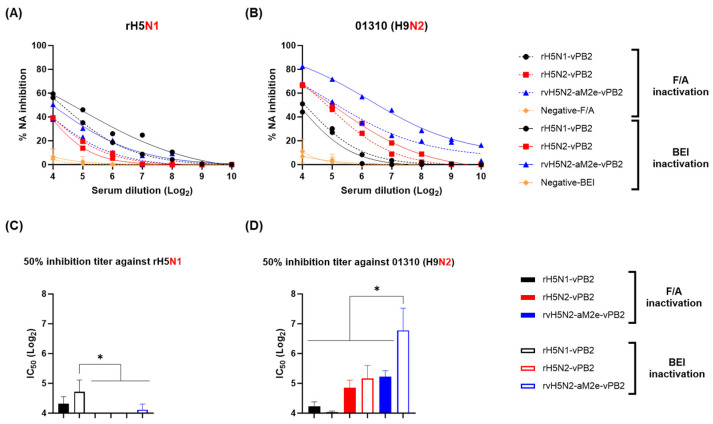
Comparison of the serum NI titers induced by recombinant virus vaccines inactivated by formaldehyde (F/A) or binary ethylenimine (BEI). To compare the immunogenicity against NA, a neuraminidase inhibition (NI) assay was performed using the serum samples collected at week 3 post vaccination. The NA activity of the virus alone was set as 100%, and the relative reduction in the NA activity due to the serum was expressed as a percentage of the NA inhibition. (**A**) NA inhibition curves for each serum sample against rH5N1. (**B**) NA inhibition curves for each serum sample against 01310 (H9N2). The curves for the vaccine groups inactivated with F/A are shown as dashed lines, and those for the BEI-inactivated vaccine groups are shown as solid lines. The 50% inhibitory concentration (IC_50_) against (**C**) rH5N1 or (**D**) 01310 (H9N2) is represented as the serum dilution titer that achieved 50% inhibition of NA activity. IC_50_ values for the F/A-inactivated vaccine groups are shown as solid bars, and those for the BEI-inactivated vaccine groups are shown as open bars, presented as mean ± SD. Statistical significance was analyzed via one-way ANOVA, and the results are denoted by asterisks (* *p* < 0.01).

**Figure 5 vaccines-13-00022-f005:**
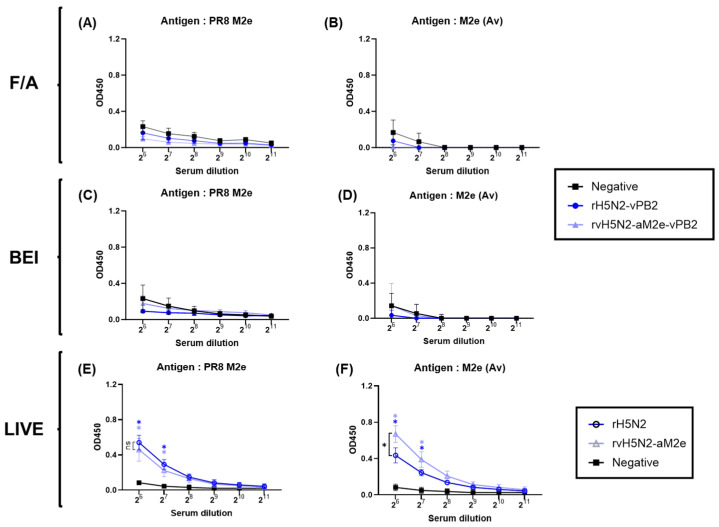
Comparison of the anti-M2e antibody levels induced by inactivated and live recombinant virus vaccines. Three weeks after inactivated vaccine administration, the serum samples collected from SPF chickens were used to evaluate the antibody responses against two distinct peptides, PR8 M2e and M2e (Av) (avian M2e), via ELISA. (**A**,**B**) IgG responses against M2e from vaccines inactivated with formaldehyde (F/A). (**C**,**D**) IgG responses against M2e from vaccines inactivated with BEI. (**E**,**F**) Antibody responses to M2e in mouse sera collected after inoculation with live viruses. Six-week-old female BALB/c mice (n = 5) were inoculated with 10^4^ EID_50_ of two live viruses (rH5N2 and rvH5N2-aM2e) or a negative control (PBS). Two weeks post-inoculation, we evaluated the IgG responses against PR8 M2e and M2e (Av) (avian M2e) via ELISA. The data are presented as the means ± SD. Statistical significance was analyzed by two-way ANOVA and is denoted by asterisks (* *p* < 0.001. ns, not significant). The black asterisks indicate significant differences between the two vaccines, whereas the blue and light blue asterisks represent significant differences between the vaccines and the negative control.

**Figure 6 vaccines-13-00022-f006:**
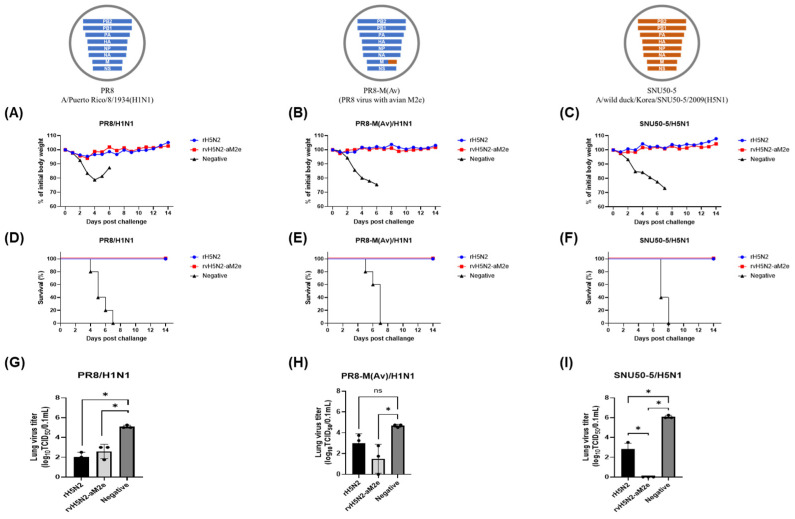
Evaluation of the weight changes, survival rates, and lung viral titers in mice inoculated with rH5N2 and challenged with multiple viruses. Five 6-week-old female BALB/c mice inoculated with 10^4^ EID_50_ of two live viruses (rH5N2, rvH5N2-aM2e) or PBS (negative) were intranasally challenged with 10^6^ EID_50_ of the SNU50-5 (A/wild duck/Korea/SNU50-5/2009 (H5N1)), PR8 (A/Puerto Rico/8/1934 (H1N1)), or PR8-M (Av) (PR8 virus with avian M2e) virus at 2 weeks after inoculation. The genome composition of each challenge virus is illustrated, with the corresponding graph displayed at the top. The color coding represents the origin of each genome segment: blue indicates segments derived from the PR8 virus, while orange represents segments derived from the SNU50-5 virus, including the avian M2e sequence. Notably, the SNU50-5 virus incorporates the avian M2e sequence. Body weight changes (**A**–**C**) and survival rates (**D**–**F**) were monitored for 2 weeks after being challenged. Three days post-challenge, the mice (n = 3) were sacrificed, and the lung viral titer (**G**–**I**) was determined. The lung viral titers are presented as the means ± SD and were analyzed via one-way ANOVA (* *p* < 0.05. ns, not significant).

**Table 1 vaccines-13-00022-t001:** Genome constellation and replication efficiency of the recombinant virus used in this study.

Recombinant Virus	HA	NA	PB2	M	PB1	PA	NP	NS	EID_50_/mL ^a^
rH5N1	K10-483/ASGR ^†^	K10-483	PR8	PR8	PR8	PR8	PR8	PR8	8.25 ± 0.47
rH5N1-vPB2	K10-483/ASGR ^†^	K10-483	01310	PR8	PR8	PR8	PR8	PR8	9.33 ± 0.23 *
rH5N2	K10-483/ASGR ^†^	01310 E20 ^#^	PR8	PR8	PR8	PR8	PR8	PR8	9.75 ± 0.4 *
rH5N2-vPB2	K10-483/ASGR ^†^	01310 E20	01310	PR8	PR8	PR8	PR8	PR8	9.0 ± 0.35
rvH5N2-aM2e	K10-483/ASGR ^†^/TGT ^‡^	01310 E20	PR8	PR8 (aM2e) ^§^	PR8	PR8	PR8	PR8	9.83 ± 0.31 *
rvH5N2-aM2e-vPB2	K10-483/ASGR ^†^/TGT ^‡^	01310 E20	01310	PR8 (aM2e) ^§^	PR8	PR8	PR8	PR8	9.58 ± 0.23 *

* Significantly different from the rH5N1 group value (*p* < 0.05). ^a^ Titer of recombinant virus reproduction in ECEs, 50% Egg Infective Dose per mL (EID_50_/mL). The data are presented as the average and standard deviation (SD) of three independent replicate experiments. ^#^ NA gene of A/Chicken/Korea/01310/2001 (H9N2) after 20 passages in embryonated chicken eggs (ECEs). ^†^ Attenuated HA cleavage site of K10-483. ^‡^ Deleted potential N-glycosylation site in the HA2 stem region. ^§^ M2e portion of the M gene from the PR8 strain was specifically modified by selecting a consensus sequence from the M2e sequence of avian influenza viruses without any potential N-glycosylation site (Appendix A).

**Table 2 vaccines-13-00022-t002:** Numbers of predicted CD8^+^ T-cell epitopes shared by vaccine strains and challenge strains.

Matching	Number of Identical Predicted CD8^+^ T-Cell Epitopes					
	HA	NA	NP	M1	NEP	Total
PR8 vs. vaccines ^a^	4	0	6	4	3	17
SNU50-5 vs. vaccines	9	0	4	3	1	17

^a^ Vaccines: rH5N2 and rH5N2-aM2e.

## Data Availability

The data presented in this study are available in this article and the Appendix A.

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
