# Peer review of "A Model H5N2 Vaccine Strain for Dual Protection Against H5N1 and H9N2 Avian Influenza Viruses"

_vaccines, 2024, doi:10.3390/vaccines13010022_

Round 1
Reviewer 1 Report
Comments and Suggestions for Authors
This study aimed to develop a model H5N2 vaccine strain for dual protection against H5N1 and H9N2 avian influenza viruses (AIVs). The authors generated PR8-derived recombinant H5N2 vaccine strains with different combinations of genes from clade 2.3.2.1c H5N1 and Y439-like H9N2 viruses, and evaluated their replication efficiency, immunogenicity, and antigenicity after inactivation with different reagents. The results demonstrated that the rvH5N2-aM2e-vPB2 strain, with modified HA2, NA, M2e, and PB2 genes, showed high productivity in embryonated chicken eggs (ECEs), and BEI inactivation significantly enhanced the immune response against NA, resulting in increased virus neutralization titers for dual protection. The study also showed that while the immune response against M2e is challenging to induce with an inactivated vaccine, it can be effectively elicited via a live vaccine.
Some major comments are suggested to further improve the manuscript.
1.The research theme of this article is not very clear. Please confirm whether the ultimate purpose of this vaccine research is for poultry or for mammals. If your vaccine is intended for the prevention and control of avian influenza in poultry, then the target animals for animal experiments should be poultry instead of BALB/c mice.
2.Whether a vaccine is truly effective mainly requires evaluating the immune protection efficiency and detecting the virus shedding situation at different time points.
3 The number of samples at each time point of each experimental group is 5, which is a little small and lacks statistical significance.
4.In the results section, for 3.3, 3.4 and 3.5, the inactivated vaccines developed with different inactivating agents should be compared in the same group at different time points for the same vaccine, which would be more meaningful.
Some minor comments are suggested to further improve the manuscript.
1. Does BEI, a virus - inactivating agent, have the potential for production and application? Please explain the cause of choosing it in the discussion section.
2. The English article lacks the content of animal ethics.
Comments on the Quality of English LanguageQuality of English language can be further improved.
Reviewer 2 Report
Comments and Suggestions for Authors
Dear Authors,
You have done very interesting research.
This research is devoted to creation of a modern inactivated recombinant vaccine against avian highly pathogenic influenza H5Nx virus, which could protect also against H9N2 virus due to induction of neutralizing antibodies by viral neuraminidase of serotype N2. The reverse genetics were applied for construction of different recombinant viruses to find out a role of each genomic element which was modified to improve property of vaccine. A lot of theoretical and experimental works were carried out to obtain an excellent result.
The article is well written. The results of the experiments are presented in tables and figures that are supplied by detailed descriptions.
A few remarks bellow.
11. Uniform designation for milliliter should be used - ml or mL.
2. Abstract
Check the abstract carefully, please. There is a designation ‘01310 PB2’. It is not clear for readers what does ‘01310’ mean. Besides, some of the abbreviations mentioned in this section are optional, for example, ECE, BSI, F/A, which occur once here.
3. Keywords
It would be better to specify the full name for ‘binary ethyleneimine inactivation’ rather than its abbreviation as a keyword.
4. Footnote for Table 1. Please, clarify item ‘a’ in Line 288.
a – Titer of virus reproduction in ECEs, 50% Egg Infective Dose per mL (EID50/mL)
5. Carefully check, please, text in lines 297-305. It seems you have confused words ‘gene’ and ‘genome’ in following phrases:
Lines 298-299: “… sequences from K10-483, six internal genomes genes from PR8 (rH5N1) and a variant with the 01310 PB2 genome gene replacing the PR8 PB2 genome gene (rH5N1-vPB2).”
Line 304: “…compatible with clade 2.3.2.1c HA and the six internal genomes genes of PR8,..”
6. Line 548. Extra parenthesis in “…153.6 hr) days) compared…”
Reviewer 3 Report
Comments and Suggestions for Authors
Manuscript ID: vaccines-3362829
Title: A model H5N2 vaccine strain for dual protection against H5N1 and H9N2 avian influenza viruses
I have revised the manuscript; some comments should be considered by the authors, such as:-
- The authors should avoid the lengthy sentences. There are few English grammatical corrections in the manuscript.
- Lines 38-40: Rewrite this sentence
- No need to write numbers in keywords
- The literature review could be expanded to provide a more comprehensive contextualization of the study within the existing body of knowledge and to highlight the novelty of the current research.
Lines 81-88: Add the reference you relied on here.
- Lines 97-107: In the last paragraph of the introduction, please state what are the key hypothesis that were tested in this work.
- The results section requires improvements in figures presentation. The figures are missing clarity.
- Discussion: inadequate discussions are made in this part. Authors just quoted the similar results of previous studies, but lack the related discussion for these results.
- Lines 564-581: Add the references you relied on here.
- After line 615: At the end of discussion, add a paragraph on describing the limitations of this work.
- Lines 616-625: Conclusion: write the conclusion again and be more concise to the major findings and suggestions.
- Insert the correct format style for journals in the list of references.
Comments on the Quality of English LanguageThe English could be improved to more clearly express the research
Round 2
Reviewer 1 Report
Comments and Suggestions for Authors
Some minor comments are suggested to further improve the manuscript.
1、In the results section, for 3.3, 3.4 and 3.5, please confirm whether the difference analysis between different inactivators is accurately labeled and clearly define the objects of comparison.
2、There are also many formatting errors in the text, In the section "2.4 Recombinant virus titration in ECEs", it is mentioned that "Each recombinant virus was subjected to 10-fold serial dilution", but the number of dilution is not stated.It is recommended to additional dilution times.
Author Response
Comment 1 : [In the results section, for 3.3, 3.4 and 3.5, please confirm whether the difference analysis between different inactivators is accurately labeled and clearly define the objects of comparison.]
Respond 1 : [In response, we have revised these sections to enhance clarity, particularly regarding the significance analysis between vaccine groups and the differentiation between inactivators. To improve the presentation of significance analysis, we have updated the significance annotations in Figure 2 and Figure 3 (Page 8, Line 367; Page 9, Line 394). Additionally, we have clarified the comparisons and objects of analysis for the differences between inactivators. The revised content has been highlighted in blue for your convenience. Specific updates can be found on Page 9 (Lines 383–386, 387), Page 10 (Lines 408–416), and Page 11 (Lines 441, 447–448, 458, 459).]
Comment 2 : [There are also many formatting errors in the text, In the section "2.4 Recombinant virus titration in ECEs", it is mentioned that "Each recombinant virus was subjected to 10-fold serial dilution", but the number of dilution is not stated.It is recommended to additional dilution times.]
Respond 2 : [Thank you for pointing out the formatting errors and suggesting the inclusion of additional dilution times in the section "2.4 Recombinant virus titration in ECEs." In response to your comment, we have revised the sentence to explicitly state the range of dilutions. The original sentence, "Each recombinant virus was subjected to 10-fold serial dilution," has been updated to "Each recombinant virus was serially diluted 10-fold using PBS, starting from a 106 -fold dilution to a 109 -fold dilution" This change has been made to ensure clarity and accuracy. The revised content can be found on Page 4, Line 189.]
Reviewer 3 Report
Comments and Suggestions for Authors
I found all the required comments were checked and revised carefully by the authors.
Comments on the Quality of English LanguageThe English could be improved to more clearly express the research.
Author Response
Comment 1 : [I found all the required comments were checked and revised carefully by the authors.]
Respond 1 : [We appreciate your acknowledgment that all the required comments were carefully checked and revised.]
Comment 2 : [The English could be improved to more clearly express the research.]
Respond 2 : [Thank you for your suggestion to improve the clarity of the English language to better express the research. We have revised parts of the results sections 3.3, 3.4, and 3.5 to enhance clarity and ensure that the findings are more effectively communicated. The revised sections have been highlighted in blue.]